# Prediabetes risk classification algorithm *via* carotid bodies and K-means clustering technique

Rafael F. Pinheiro, Maria P. Guarino, Marlene Lages and Rui Fonseca-Pinto

Center for Innovative Care and Health Technology (ciTechCare), School of Health Sciences (ESSLei), Polytechnic University of Leiria, Leiria, Leiria, Portugal



## ABSTRACT

Diabetes is a disease that affects millions of people in the world and its early screening prevents serious health problems, also providing relief in the demand for healthcare services. In the search for methods to support early diagnosis, this article introduces a novel prediabetes risk classification algorithm (PRCA) for type-2 diabetes mellitus (T2DM), utilizing the chemosensitivity of carotid bodies (CB) and K-means clustering technique from the field of machine learning. Heart rate (HR) and respiratory rate (RR) data from eight volunteers with prediabetes and 25 without prediabetes were analyzed. Data were collected in basal conditions and after stimulation of the CBs by inhalation of 100% of oxygen and after ingestion of a standardized meal. During the analysis, a greater variability of groups was observed in people with prediabetes compared to the control group, particularly after inhalation of oxygen. The algorithm developed from these results showed an accuracy of 86% in classifying for prediabetes. This approach, centered on CB chemosensitivity deregulation in early disease stages, offers a nuanced detection method beyond conventional techniques. Moreover, the adaptable algorithm and clustering methodology hold promise as risk classifications for other diseases. Future endeavors aim to validate the algorithm through longitudinal studies tracking disease development among volunteers and expand the study's scope to include a larger participant pool.

## INTRODUCTION

Diabetes is a metabolic disease, silent and insidious, that currently affects over 500 million people worldwide (*Sun et al., 2022*). Although the first record of diabetes was made hundreds of years before Christ (*Stapley, 2001*), this disease is still responsible for thousands of deaths every year. Its early detection can prevent serious damage to health and decrease the risk of death (*Fitriyani et al., 2019*).

In search of developing tools that can help in the classification or early diagnosis of diabetes, many works have been done in the area of machine learning, some in a non-invasive way (*Ahmed et al., 2022*; *Chaki et al., 2022*). Still, within this area, several techniques have been used, which also use data obtained invasively, for example: with

Corresponding author
Rafael F. Pinheiro,
rafael.f.pinheiro@ipleiria.pt

support vector machine (SVM) (*Patil et al., 2022*); K-means algorithm (*Zhu, Idemudia & Feng, 2019*); artificial neural networks (*Parab et al., 2021*); and deep learning (*Zhu et al., 2021*; *Khan et al., 2024*). In addition, much work for diabetes classification involves the construction of hybrid algorithms (*Zarkogianni et al., 2018*; *Saleh & Brixtone Batou, 2022*; *Zhu et al., 2023*; *Anbananthen et al., 2023*).

More specifically on the diabetes risk problem, many studies have emerged in the literature that utilize approaches involving machine and deep learning architectures. For instance, in *Ramesh, Aburukba & Sagahyroon (2021)*, an end-to-end remote monitoring framework is proposed for automated diabetes risk classification using SVM, alongside management facilitated by personal health devices, smart wearables, and smartphones. A different architecture is presented by *Nilashi et al. (2023)*, which uses deep learning combined with singular value decomposition and self-organizing maps to predict the risk of diabetes mellitus. Additionally, *Li et al. (2023)* integrates methods of deep learning and logistic regression to develop a risk model for type-2 diabetes mellitus (T2DM). Still more recently, *Bülbül (2024)* introduces a hybrid deep learning model that combines a genetic algorithm, a stacked autoencoder, and a Softmax classifier. These diverse approaches highlight the topic's peak and the ongoing advancements in utilizing machine and deep learning techniques for effective diabetes risk classification.

For the characterization, diagnosis and classification of diseases, the literature describes several biomarkers that have a broad definition. Herein, a biomarker is assumed as a mensurable biological feature that indicates an ongoing pathogenic process. One biomarker for the detection of prediabetes is the carotid bodies (CB) function assessment (*Conde et al., 2022*). CBs are small chemoreceptor organs that function as metabolic sensors. They are located at the bifurcation of the common carotid artery bilaterally and measure 5–7 mm in length. An increased activity of CBs in individuals with prediabetes has been shown *via* the Dejours Test (*Dejours, 1962*; *Conde et al., 2022*), a challenge-test that uses a transient period of 100% O2 inhalation (hyperoxia) as a stimulus, to indirect assessment of CB activity through changes in cardiorespiratory response (*Cunha-Guimaraes et al., 2020*).

According to the American Diabetes Association, the criteria for diagnosing diabetes are: fasting plasma glucose $\geq 126$ mg/dL OR $2-h$ postprandial glucose $\geq 200$ mg/dL during oral glucose tolerance test (OGTT) OR HbA1C $\geq 6.5\%$ OR in a patient with classic symptoms of hyperglycemia or hyperglycemic crisis, a random plasma glucose $\geq 200$ mg/dL. The OGTT, although widely considered the gold standard dynamic test for diagnosing diabetes and prediabetes (*Chamberlain et al., 2016*; *Jagannathan et al., 2020*), as well as fasting glucose, have certain limitations. Older studies, such as *Gross et al. (2002)*, already pointed out failures in detecting gestational diabetes and prediabetes. Similarly, *Carnevale-Schianca et al. (2003)* highlights that reliable diagnoses and clinical decisions cannot be based solely on fasting glucose, emphasizing the need for a more comprehensive diagnostic approach. More recent research, such as *Bogdanet et al. (2020)*, also identifies the limitations of OGTT during pregnancy. Additionally, *Lages et al. (2022)* expands on this discussion by recognizing that OGTT, besides not reflecting people's usual eating

patterns, can cause adverse effects such as vomiting, diarrhea, bloating, and a risk of hypoglycemia and other associated complications.

In order to explore alternatives to current early diagnosis tools for diabetes, the CBmeter prototype was developed (*Lages et al., 2021*) to test the hypothesis that indirect assessment of CB activity may detect metabolic diseases in an early phase. The CBmeter is a device that aims to collect and analyze data on heart rate (HR), respiratory rate (RR), oxygen saturation and interstitial glucose data in real time and synchronously to relate the function of CBs to metabolic diseases features. This test can provide an early indication of prediabetes by assessing carotid body chemosensitivity, which correlates with insulin resistance independently of fasting glucose levels (*Cunha-Guimaraes et al., 2020*; *Conde et al., 2022*). The advantage of the CBmeter is its ability to detect early metabolic changes without invasive procedures, offering a potential tool for early diagnosis before fasting glucose or other traditional biomarkers indicate pathology. There are other publications regarding the development of the CBmeter: *Lages et al. (2021)* sets out the essential protocols for research and development of the equipment; *Brito et al. (2018)* presents the software for signal acquisition and processing; *Fonseca-Pinto et al. (2020)* introduces a methodology for indicating metabolic disease from the machine learning technique known as principal component analysis (PCA); *Conde et al. (2022)* establishes CB as a new biomarker for the early detection of prediabetes.

As a sequence of the CBmeter development work, the objective of this article is to present a new hybrid algorithm, which performs clustering of CBmeter data *via* K-means technique of machine learning and a scoring technique developed in this work. The algorithm provides an estimate of risk (high, medium, or low) for a given patient to enter the prediabetes group based on data from the CBmeter with oxygen stimuli and a meal. The methodology presented herein consists in the formation of clusters of data from volunteers with prediabetes and volunteers without prediabetes (control volunteers) *via* K-means algorithm using data of time in minutes, HR and RR obtained by data collection with the CBmeter. Through these clusters, the data are analyzed and strategies are established for the creation of the classification algorithm. To facilitate the construction of computational codes, for example in python or Matlab, the article provides three pseudo-algorithms that summarize all the procedures presented in this work.

The main contributions of this article can be listed as follows:

- According to the authors' literature search, there are no previous works that deal with the development of algorithms using K-means and CB for the classification of risk for prediabetes. Therefore, this work ensures originality and presents a new methodology in the development of risk classification algorithms. Furthermore, the theory can also be applied to other pathologies.
- The methodology developed allows for a clear differentiation in the characterization of the clustering patterns of patients with prediabetes and control. A new variable was inserted, namely variability, which enables to verify the difference between the clusters of volunteers with prediabetes and control.

| Algorithm 1   K-means for time and HR, RR and RR×HR variables. |
| --- |

**Result:** Matrix $CPOX_{g,var}$ of clusters *via* K-means algorithm of volunteers of group $g$ and variable *var* of the 3 min after inhalation of oxygen.

**Input:** Patient matrix $P_{g,var}$ containing the time, group and variable. Where:

$g \leftarrow 1 = prediabetes \; or \; 2 = control$

$var \leftarrow 1 = HR, \; 2 = RR \; or \; 3 = RR \times HR$

**FOR** $i \leftarrow 1 :$ last patient

$patient \leftarrow P_{g,var}(0, i)$ %Selects time column 0 and patient $i$ to perform the cluster%

$patient \leftarrow scaler(patient)$ Standardizes the 0 and $i$ columns to the same scale.

$KmeansP \leftarrow Kmeans(cluster \; number, patient)$ %Applying the

K-means algorithm%

$CP_{g,var}(0, i) \leftarrow KmeansP$ %Saves the labels obtained from the patient clusters $i$ %

**endFOR**

**FOR** $t \leftarrow 11 : 13$

$CPOX_{g,var}(t, :) \leftarrow CP_{g,var}(t, :)$ %Stores in the CPOX matrix the lines of%

%time 11, 12 and 13 which correspond 3 min after oxygen inhalation%

- The prediabetes risk classification algorithm (PRCA) training process, which includes the development of the scoring matrix ($\Psi$), which relates the clusters, the variables (HR, RR and RR×RH) and the glucose level (fasting and postprandial) of the volunteers, is part of an innovative structure. From $\Psi$, a new variable is derived, called maximum risk ($\chi$). The novelty does not reside in the matrices *per se*, but in the whole approach, including the construction of $\Psi$ and $\chi$, as formally established by Eqs. (2)–(4) and Algorithm 2, all of which are original contributions.

- The approach used is a non-invasive diagnostic support methodology. After obtaining the $\chi$ value during the training process with volunteers with prediabetes and their respective glucose values (fasting and postprandial), for new indications of prediabetes risk, only the HR and RR data of patients undergoing the CBmeter protocol, which includes inhaling $O_2$ and eating a specific meal, are needed.

- Algorithm 3 completes the PRCA, which provides an indication of risk for a given patient to be part of the group of people with prediabetes. The algorithm is original and uses Algorithms 1 and 2 for its operation and construction. Furthermore, it is easy to adapt for application in other pathologies, with larger numbers of variables, volunteers, and clusters.

The content of this article is not trivial. However, an effort has been made to write it in a language that makes it easy for researchers in the fields of health, medicine and engineering to understand. In addition, source code, raw data and a README.txt are provided as Supplemental Files to guide researchers interested in reproducing the results. Finally, the rest of the article is structured as follows: In the next section, one has the Methods; in third section, the results are presented; in the fourth section, a discussion of some important

---

**Algorithm 2  Maximum risk.**

---

**Result:** Maximum Risk ($\chi$).

**Input:** Matrix $G$ with the glucose level of the volunteers with prediabetes at each minute, where the lines are the time and the columns the glucose level of each volunteer; Matrix $CPOX_{g,var}$ from Algorthm 1; Vector $ID$ with the identification of the volunteers; Number of cluster names $n$; Number of variables $m$.

**Calculate** Severity Level (SL) and Weight (W) by Eqs. (2) and (3).

**FOR** $i \leftarrow 1$ : last patient

      **FOR** $j \leftarrow 1$ : last minute

          $SL(i) \leftarrow equation\_2[G(i,j)]$

   $W(i) \leftarrow equation\_3[SL(i)]$

**endFOR**

**Convert** Matrix $CPOX_{g,var}$ to matrix $N_{g,var}$ with the cluster names and with the attribution of the values $[1, 2, 3, 4, 5]$ to the clusters DU, DD, DUD, S, T, respectively.

**Calculate** Association Matrix $\Delta$.

$\Delta \leftarrow [ID, W, N_{1,1}, N_{1,2}, N_{1,3}]$

**Calculate** Score Matrix ($\Psi$).

**FOR** $j \leftarrow 1 : m$

  **FOR** $i \leftarrow 1 : n$

    $sum \leftarrow 0$

    **FOR** $r \leftarrow 1 : n$

      **IF** $\Delta(r, j+1) = i$

        $sum \leftarrow sum + \Delta(r, 1)$

      **endFOR**

    $\Psi(i, j) \leftarrow sum$

  **endFOR**

**endFOR**

**Calculate** Maximum Risk ($\chi$) by Eq. (4).

$sum \leftarrow 0$

**FOR** $j \leftarrow 1 : m$

  $max \leftarrow 0$

  **FOR** $i \leftarrow 1 : n$

    **IF** $\Psi(i, j) > max$

      $max \leftarrow \Psi(i, j)$

  **endFOR**

  $sum \leftarrow sum + max$

**endFOR**

$\chi \leftarrow sum$

---

| Algorithm 3 | Prediabetes Risk Classification algorithm (PRCA). |
|---|---|

**Result:** Return "Low Risk", "Medium Risk" or "High Risk".

**Input:** Matrix $D$ with patient data for prediabetes risk analysis; score matrix $\Psi$; maximum risk number ($\chi$); number of cluster names $n$; number of variables $m$.

**K-means:** Apply Algorithm 1 to obtain the matrix $D_{OX}$ (adapt Algorithm 1 to receive only one patient.).

**Convert** $D_{OX}$ to $N_{OX}$.

**Calculate** The score:

$score \leftarrow 0$

**FOR** $j \leftarrow 2 : 1 + m$

  **FOR** $i \leftarrow i : n$

    **IF** $N_{OX}(j) = \Psi(i, 1)$

      $score \leftarrow score + \Psi(i, j)$

  **endFOR**

**endFOR**

**Get** The risk:

**IF** $score < \frac{\chi}{3}$

  **Return:** "Low Risk"

**IF** $\frac{\chi}{3} \leq score < 2\frac{\chi}{3}$

  **Return:** "Medium Risk"

**IF** $2\frac{\chi}{3} \leq score \leq \chi$

  **Return:** "High Risk"

aspects regarding the work is done; and the last section closes the article with the conclusions.

# METHODS

## Participants and data collection

The participants were selected according to the methodology described in the CBmeter study protocol, approved on 10 January 2019 by the Ethics Committee of the Leiria Hospital Centre. Volunteers with prediabetes or T2DM and volunteers without prediabetes were recruited in a multicentric, non-randomized controlled interventional study conducted in Portugal during 18 months, resulting in the inclusion of eight volunteers with prediabetes and 25 healthy controls. Data from eight volunteers with prediabetes and 25 volunteers without prediabetes (control volunteers) were collected according to the protocol previously mentioned. Briefly, the volunteers were assessed for the variables of interest in baseline conditions during 10 min and afterwards, submitted to 100% oxygen inhalation during 10 s, and given a standardized meal after the twentieth minute (21). HR and RR data collection was performed by CBmeter *via* CBView software (*Brito et al., 2018*), a real-time physiological signal acquisition and processing system at an

acquisition rate of 500 Hz. Interstitial glucose (iGlu) was collected every minute by means of a flash glucose monitoring system (FreeStyle Libre, Abbott, Lake County, IL, USA). The study was conducted in accordance with the Declaration of Helsinki, been previously approved by the Ethics Committee of the Leiria Hospital Centre (Protocol number PI.NC. EC.2018.01). Written informed consent was obtained *via* a specific form from all the participants and/or their legal guardians. The full protocol of the research can be seen at *Lages et al. (2021)*.

## Data dimensionality

As indicated in the previous subsection, the data comprises two groups of voluntaries: patients with prediabetes and patients control. The algorithms created use time in the clustering processes. Thus, the data used comprises time series ranging from 1 to 80 min, with samples taken every minute for the HR, RR and RR×HR variables. It is therefore important to note that the dataset is not two-dimensional, *i.e.*, *patients × variables*, but three-dimensional, that is, *patients × variables × time*. In formal mathematical notation: $\mathbf{X} \subset \mathbb{R}^{N \times V \times T}$ where:

- $N$ represents the number of patients,
- $V$ represents the number of variables,
- $T$ represents time.

Therefore, the dataset is described by:

$$\mathbf{X} = \left(\mathbf{X}_{i,v,t}\right) \text{ for } i \in \{1, 2, \ldots, N\}, v \in \{1, 2, \ldots, V\}, t \in \{1, 2, \ldots, T\}.$$

In this work, $N$ can be 8 (group of people with prediabetes) or 25 (control group), $V$ is 3 (for the HR, RR and RR×HR variables) and $T$ is 80 (time ranges from minute 1 to minute 80). This characteristic of multidimensionality and its computational operationalization can be clearly seen in Algorithm 1, presented in the Results section.

## The K-means algorithm

To perform the clustering, the K-means algorithm from the Python library sklearn was used. This algorithm is based on unsupervised learning, which learns the relationships between data identifying when a group begins and ends another through a mathematical metric (*Hartigan & Wong, 1979*; *Bock, 2007*). In its traditional version, the K-means algorithm has the number K of clusters as input. From that, there is a random initialization of centroids. Then, for each point in the database, a calculation of the distance from the point to the centroids is performed, then this point is associated with the nearest centroid. After this, the algorithm calculates the average of all the points associated to each centroid and establishes a new centroid. The algorithm repeats the previous steps until there are no more updates of elements in groups.

For the analysis of HR and RR, the K-means method stands out for its simplicity and computational efficiency. K-means is particularly suitable when the number of clusters can

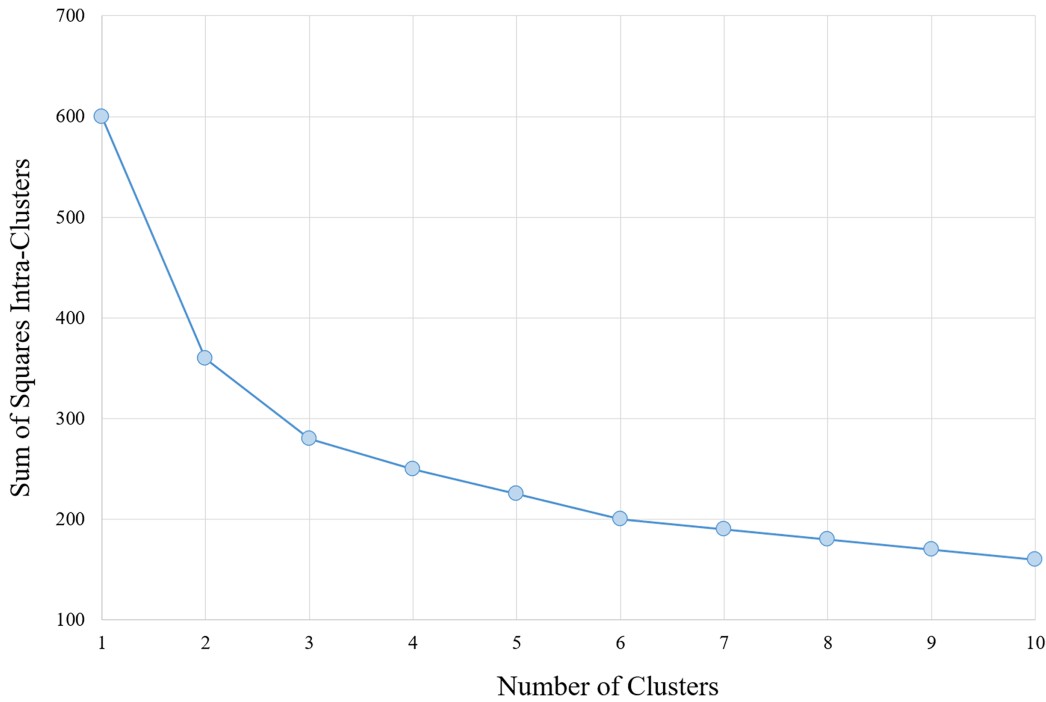

**Figure 1 WCSS graph indicating the appropriate number of clusters for the K-means algorithm.**

be adequately defined, as will be shown later in the article, providing a clear segmentation of the different physiological states based on the patterns observed in the biosignals. Unlike complex methods such as Gaussian mixture models (GMM) and spectral clustering (*Srivastava, Sarkar & Hanasusanto, 2023*), which require intensive parameter adjustments and can be sensitive to the initial choice of parameters, K-means allows for straightforward implementation and intuitive interpretation of clustering results. Compared to DBSCAN (*Bushra & Yi, 2021*), known for its robustness in dealing with variable densities and outliers, K-means is more appropriate when a clear segmentation of the data into defined clusters is desired. While approaches based on deep learning, such as Autoencoders (*Guo et al., 2017*), offer the capacity to capture non-linear complexities, they are more computationally complex and less straightforward in terms of interpretability. Thus, K-means remains a viable choice for biosignal analysis, standing out for its practical applicability and ease of use.

## Number of clusters

The determination of the number of clusters to be used in the classification algorithm is based on analysis of the results of the section Analysis, arriving at a number of six clusters for the data with the characteristics provided. Moreover, corroborates with this number, analysis performed *via* algorithm within-cluster sum-of-squares (WCSS), which measures the variation within each cluster. The WCSS technique establishes that the most appropriate number of clusters for a given sample is obtained when there is less variation in the decreasing curve. An example can be verified in Fig. 1.

## Data augmentation technique

As the samples are unbalanced (eight volunteers with prediabetes and 25 without), data augmentation techniques are needed for more reliable validation. In this work, the adaptive synthetic sampling approach for unbalanced learning (ADASYN) (*He et al., 2008*) was used, which is a variant of the synthetic minority over-sampling technique (SMOTE) (*Chawla et al., 2002*). This method was implemented using MATLAB® R2020a (The MathWorks, Inc., Natick, MA, USA).

The synthetic minority oversampling technique (SMOTE) is a method used to address class imbalance in machine learning datasets by generating synthetic instances of the minority class. Rather than simply duplicating existing minority examples, SMOTE creates new, plausible samples by interpolating between existing data points. It selects a minority instance, identifies its nearest neighbors from the same class, and generates synthetic samples along the line segments connecting the instance to its neighbors. This approach increases the diversity of the minority class and helps improve model performance, particularly when the imbalance between classes leads to poor learning of minority patterns.

On the other hand, adaptive synthetic sampling (ADASYN) extends SMOTE by introducing an adaptive component that focuses on the most challenging examples in the minority class. Instead of creating samples uniformly, ADASYN generates more synthetic examples in regions where the minority class is harder to distinguish from the majority class. This targeted approach enhances the model's ability to learn from difficult areas, refining the decision boundary and reducing bias in favor of the majority class.

## RESULTS

The results of this article comprise two main parts: the first main part is made up of subsections "Clustering process and analysis", "Training process: score matrix and maximum risk number", and "Prediabetes risk classification algorithm (PRCA)". These subsections deal with the development of the PRCA, including some analysis, presentation of Algorithms 1–3 and examples. The second part comprises the subsections "Performance and Validation" and "Comparison with other methods", aiming to show the validation process using the four-fold Cross-Validation technique, while also highlighting the PRCA relevance through comparisons with other studies employing cutting-edge technologies.

In support of this structure, Fig. 2 illustrates the complete framework of the PRCA, outlining the primary functions of each part and providing an overview of the algorithm. In the figure, Algorithm 1 consists of clustering the data *via* K-means, which makes it possible to identify patterns in the HR and RR data clusters of subjects submitted to the CBmeter protocol (inhalation of $O_2$ and ingestion of a specific meal). The function of Algorithm 2 is to train the PRCA using the clusters provided by Algorithm 1 and glucose data from the volunteers selected for training. Finally, Algorithm 3 connects Algorithms 1, 2 and the HR and RR data of new patients (submitted to the CBmeter protocol) for risk analysis and classification. It is important to note that Algorithm 3 does not require glucose information for the subjects under analysis, thus making PRCA a non-invasive method for supporting the diagnosis of prediabetes.

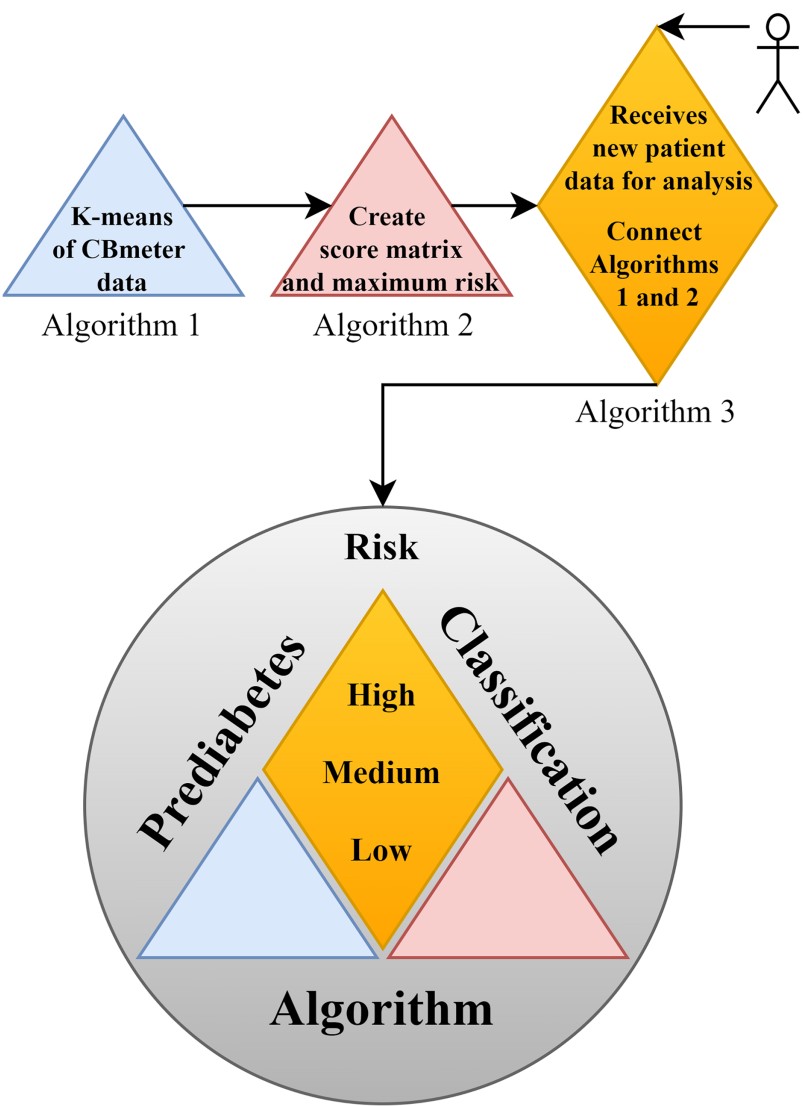

**Figure 2 Schematic diagram of the prediabetes risk classification algorithm (PRCA).**

## Clustering process and analysis

In this first part of the results, using the K-means clustering technique, clusters analyses were performed to identify patterns and determine strategies for the construction of the classification algorithm, as well as for the determination of the most adequate number of clusters. The variables (features) under analysis are time, HR, RR and RR×HR. These were considered within the groups of people with prediabetes and control patients.

The clusters were generated using Algorithm 1, which receives a matrix $P_{g,var}$ with the data of a given variable for all volunteers of a given group. Since there were two groups (individuals with prediabetes and control) and three variables (HR, RR and RR×HR), it was assigned the following values: 1 for the group of people with prediabetes and 2 for the control group; 1 for the HR variable, 2 for RR and 3 for RR×HR. For example, for $P_{1,1}$ one has the matrix of the group of patients with prediabetes with HR data, where the first

**Table 1 Form of presentation (matrix with 80 rows and nine columns) of HR data of volunteers with prediabetes for the Algorithm 1, where the time is given in minutes, and the HR in beats per minute.**

| TIME | HR1 | HR2 | HR3 | ... | HR8 |
|------|-----|-----|-----|-----|-----|
| 1 | 59 | 68 | 81 | ... | 63 |
| 2 | 59 | 67 | 80 | ... | 65 |
| 3 | 57 | 63 | 81 | ... | 62 |
| ⋮ | ⋮ | ⋮ | ⋮ | ⋱ | ⋮ |
| 80 | 62 | 69 | 90 | ... | 69 |

column of the matrix is the time in minutes (from 1 to 80) and the following columns are the HR of each patient with prediabetes (eight patients). So, in this example, one has a matrix with 80 lines and nine columns, that is, $P_{1,1} \in \mathbb{R}^{80 \times 9}$ (see Table 1). In the case of the variable RR×HR from the group of patients with prediabetes, the matrix will be $P_{1,3} \in \mathbb{R}^{80 \times 17}$ (see Table 2). It is important to highlight that the k-means clustering algorithm runs using HR, RR and RR×HR during the entire time interval, *i.e.*, from minute 1 to minute 80.

Algorithm 1 returns the labels of the clusters between the time and the variable of each volunteer in the matrix $CPOX_{g,var}$ already in the 3 min after 100% oxygen administration, that is, times 11, 12 and 13 min. For this work, the term "variability" was defined as the index in percentage of the variation between the clusters of the volunteers of each group and each variable. The formula for calculating variability is given by Eq. (1). A lower index of variability means that in each group, for a given variable, a higher number of a given pattern of equal clusters was observed. In the HR analysis, this account is given in detail in order to clarify the concept of variability presented in this work.

$$Variability = \frac{C_d}{N_s} \times 100,$$
(1)

where $C_d$ represents the total number of clusters of different colors for the same variable and the same types of volunteers, and $N_s$ denotes the total number of volunteers in the same group (people with prediabetes or control) for the same variable.

### Dejours test and time analysis

CB chemosensitivity was evaluated by two breaths of 100%O2, the double-breath Dejours test, as with this test is possible to obtain a change in the oxygen drive almost free of secondary factors since they are secondary in time (*Dejours, 1962*). To assess peripheral chemosensitivity respiratory was measured using the CBmeter, since it was already demonstrated that increased CB chemosensitivity observed in patients with prediabetes is seen in the RR, but not in tidal volume (*Cunha-Guimaraes et al., 2020*).

Thus, RR was assessed with the CBmeter while the subjects breathed room air (21% O2; normoxia), followed by two breaths of 100% O2 (hyperoxia) delivered at a 10 L/min flow, and then normoxia again. Hyperoxia applied during a few seconds resulted in a decrease in

**Table 2 Form of presentation (matrix with 80 rows and 17 columns) of RR×HR data of volunteers with prediabetes for the Algorithm 1, where the time is given in minutes, and the HR and RR in beats per minute.**

| TIME | RR×HR1 | | RR×HR2 | | RR×HR3 | | ... | RR×HR8 | |
|------|--------|------|--------|------|--------|------|-----|--------|------|
| 1 | 6 | 59 | 15 | 68 | 15 | 81 | ... | 18 | 63 |
| 2 | 8 | 59 | 19 | 67 | 16 | 80 | ... | 18 | 65 |
| 3 | 15 | 57 | 15 | 63 | 17 | 81 | ... | 17 | 62 |
| ⋮ | ⋮ | ⋮ | ⋮ | ⋮ | ⋮ | ⋮ | ⋱ | ⋮ | ⋮ |
| 80 | 16 | 62 | 17 | 69 | 17 | 90 | ... | 18 | 69 |

ventilation that reflects CB chemosensitivity exclusively, without interference of the central nervous system chemoceptors (*Honoring, 2003*). The maneuver was repeated three times in each patient to assess reproducibility of the test. If 100% O2 is breathed during prolonged exposures (several minutes), ventilation does not change or could even have increased.

The magnitude of the ventilatory depression caused by hyperoxia, often used as an index of carotid body sensitivity although without definition of a gold standard protocol. Herein, it was assessed the ventilatory responses to a brief hyperoxic test of 10 s to avoid interference of the central nervous system chemoreceptors. RR was monitored before and during the hyperoxic test with the CBmeter. Figure 3 shows that the mean of the RR of volunteers with prediabetes, in general are lower than those of control volunteers. Furthermore, by the Independent-Samples Mann-Whitney U Test ($p = 0.007$), it was concluded that the first 3 min after hyperoxia (11, 12, and 13) are linked to a greater ventilation decrease in response to the hyperoxic test (see red line in Fig. 3).

Therefore, this analysis verifies significant RR differences at times 11, 12 and 13. In the following analyses, *via* the K-means algorithm, the term "variability" is designated to name this pattern.

### HR analysis

Basically, it is each person's data that is clustered by the Algorithm 1. For example, the HR data and time of a volunteer with prediabetes is submitted to the K-means algorithm with a K value of 6. It should be noted that the method consists of clustering Time and HR; Time and RR; and Time and RR×HR. This procedure can be checked in detail in "# OBTAINS THE CLUSTERS FOR K=6" of the Python code "code_clusters_CBmeter.txt" provided as Supplemental Material. The algorithm returns these data separated and labeled into six groups. Figure 4 is a result of Algorithm 1 for volunteers with prediabetes in the HR data.

Looking only at the first three columns of Fig. 4, HR1 is the column in which the HR data is recorded over the 80-min Time (1st column) of subject 1. As the number of clusters chosen for K-means was 6, the CL1 column shows how the HR1 data was clustered over time by K-means. Figure 4 shows that the Time 11 and 12 min and HR1 data of 58 were placed in group 2 by K-means. The pair of minutes 13 and HR1 of 62 was placed in group 5. This process is carried out for each minute (between 1 and 80), where for each pair

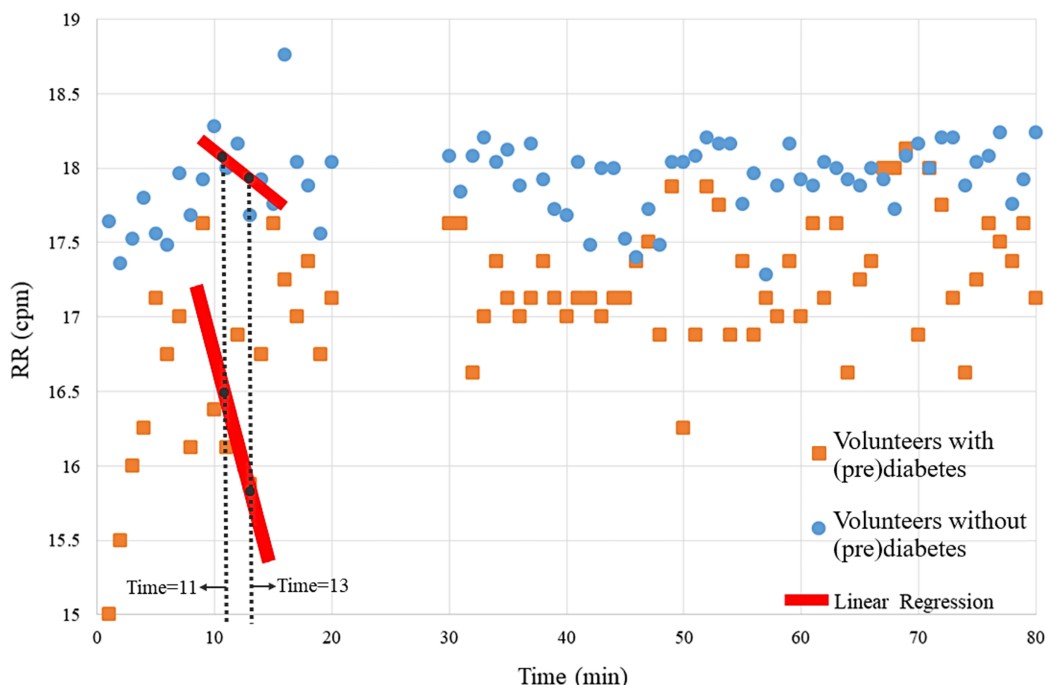

**Figure 3 Means of the volunteers' RR over time.** Each point on the graph represents the mean RR data from volunteers with and without prediabetes at each minute. For example, at time 20, the RR values of volunteers without prediabetes were summed and divided by the total number of volunteers without prediabetes, resulting in approximately 17.1 breaths per minute.

| Time (min) | HR1 | CL1 | HR2 | CL2 | HR3 | CL3 | HR4 | CL4 |
|---|---|---|---|---|---|---|---|---|
| 11 | 58 | 2 | 66 | 1 | 80 | 3 | 75 | 5 |
| 12 | 58 | 2 | 66 | 1 | 82 | 3 | 75 | 5 |
| 13 | 62 | 5 | 63 | 5 | 85 | 0 | 76 | 5 |

| Time (min) | HR5 | CL5 | HR6 | CL6 | HR7 | CL6 | HR8 | CL8 |
|---|---|---|---|---|---|---|---|---|
| 11 | 56 | 1 | 69 | 1 | 73 | 2 | 65 | 1 |
| 12 | 54 | 4 | 63 | 5 | 69 | 2 | 60 | 5 |
| 13 | 56 | 1 | 62 | 5 | 67 | 5 | 64 | 5 |

**Figure 4 HR clusters provided by Algorithm 1 for volunteers with prediabetes, where HR1, in beats per minute, represents the HR data of volunteer 1 with prediabetes, and CL1 represents the group (out of a total of six) in which each volunteer's HR information was placed.** The same interpretation follows for the other volunteers.

($Time_i$, $HR1_i$) K-means indicates to which group it belongs. During K-means processing, the pairs ($Time_i$, $HR1_i$), for $i = 1, 2, \ldots, 80$, can change groups until they reach the point where no more updates are possible (for example, when the centroids no change).

This method of using K-means allows to find a pattern in minutes 11, 12 and 13, which are the 3 min after inhaling oxygen, and this is the return from Algorithm 1 for the variables (HR, RR, RR×HR) of all the subjects. Also note that the HR4 data for minutes 11, 12 and 13 remain in the same group (group 5). On the other hand, for all the other volunteers, there is a variation between groups, between minutes 11, 12 and 13, which is the case for volunteer HR1. As explained before, note that for this volunteer the HR data fell into groups 2 and 5. This procedure enabled a clear observation that, in minutes 11, 12, and 13, the HR data exhibit patterns that distinguish volunteers with prediabetes from volunteers without prediabetes.

**Remark 1.** *It is important to note that the method is not real-time; it is necessary to have all the data (a complete collection of HR and RR over the 80 min) before applying Algorithm 1 and obtaining the clusters.*

In the HR analysis, grater variability is observed between minutes 11 and 13 (post-oxygenation) for the volunteers. From Fig. 4, only the fourth volunteer (all blue) of the cluster of peoples with prediabetes did not present any group variation, and seven volunteers had a variation of groups (clusters of different colors, that is, $C_d$) between minutes 11, 12 and 13. Applying Eq. (1), one has:

$$Variability = \frac{C_d}{N_s} \times 100 = \frac{7}{8} \times 100 = 87.5\%.$$

Figure 5 shows the clusters (CL) for the 25 control volunteers during minutes 11, 12 and 13. One can notice less variability for this group. That is, of the 25 volunteers, 12 had an inter-group variation which corresponds to 48% of the total, that is:

$$Variability = \frac{C_d}{N_s} \times 100 = \frac{12}{25} \times 100 = 48\%.$$

To determine the ideal number of clusters, the analysis was carried out with various numbers of clusters (K) as shown in Fig. 6, and the WCSS algorithm was applied, whose graph is shown in Fig. 1. Figure 6A shows the variability percentages for other cluster numbers. The graph has a trend line obtained by means of polynomial regression of degree 3, where the horizontal axis represents the number of clusters and the vertical axis represents the variability in percentage.

**Remark 2.** *Figures 4 and 5 show the 3 min after oxygen inhalation (11, 12 and 13), where HR_ and CL_ represent the HR and the cluster indicated by the algorithm respectively for each volunteer. Colors were added for better visualization and to distinguish the types of groups found. It should be noted that each series of clusters covers a time series between minute 1 and minute 80 for each volunteer. For example, CL1 represents the group labels returned by K-means for volunteer 1's HR data, indicating that minutes 11 and 12 of HR1 belong to group 2 (blue) and minute 13 belongs to group 5 (orange). The same analysis is carried out for RR and RR×HR.*

### RR analysis

Similarly to the HR analysis, in the RR analysis, also using $K = 6$, greater variability was observed between minutes 11 and 13 (post-oxygenation) for the volunteers with

| Time (min) | HR1 | CL1 | HR2 | CL2 | HR3 | CL3 | HR4 | CL4 | HR5 | CL5 |
|---|---|---|---|---|---|---|---|---|---|---|
| 11 | 62 | 2 | 74 | 4 | 85 | 3 | 62 | 5 | 66 | 5 |
| 12 | 58 | 2 | 78 | 4 | 71 | 4 | 57 | 1 | 61 | 3 |
| 13 | 61 | 2 | 83 | 1 | 72 | 4 | 57 | 1 | 61 | 3 |

| Time (min) | HR6 | CL6 | HR7 | CL7 | HR8 | CL8 | HR9 | CL9 | HR10 | CL10 |
|---|---|---|---|---|---|---|---|---|---|---|
| 11 | 71 | 2 | 66 | 4 | 82 | 1 | 75 | 2 | 67 | 4 |
| 12 | 75 | 3 | 62 | 4 | 83 | 1 | 75 | 2 | 65 | 4 |
| 13 | 74 | 3 | 64 | 4 | 83 | 1 | 76 | 2 | 66 | 4 |

| Time (min) | HR11 | CL11 | HR12 | CL12 | HR13 | CL13 | HR14 | CL14 | HR15 | CL15 |
|---|---|---|---|---|---|---|---|---|---|---|
| 11 | 70 | 2 | 76 | 1 | 74 | 4 | 69 | 5 | 76 | 1 |
| 12 | 76 | 5 | 73 | 5 | 74 | 4 | 68 | 5 | 78 | 5 |
| 13 | 80 | 1 | 72 | 5 | 81 | 1 | 70 | 5 | 76 | 1 |

| Time (min) | HR16 | CL16 | HR17 | CL17 | HR18 | CL18 | HR19 | CL19 | HR20 | CL20 |
|---|---|---|---|---|---|---|---|---|---|---|
| 11 | 70 | 3 | 68 | 2 | 57 | 0 | 68 | 1 | 66 | 1 |
| 12 | 73 | 3 | 68 | 2 | 58 | 0 | 65 | 1 | 66 | 1 |
| 13 | 72 | 3 | 69 | 2 | 54 | 0 | 66 | 1 | 70 | 1 |

| Time (min) | HR21 | CL21 | HR22 | CL22 | HR23 | CL23 | HR24 | CL24 | HR25 | CL25 |
|---|---|---|---|---|---|---|---|---|---|---|
| 11 | 69 | 0 | 60 | 2 | 77 | 5 | 79 | 2 | 67 | 5 |
| 12 | 75 | 2 | 63 | 2 | 81 | 3 | 72 | 4 | 69 | 0 |
| 13 | 70 | 0 | 58 | 2 | 78 | 5 | 70 | 4 | 68 | 5 |

**Figure 5 HR clusters provided by Algorithm 1 for volunteers without prediabetes, where HR1, in beats per minute, represents the HR data of volunteer 1 without prediabetes, and CL1 represents the group (out of a total of six) in which each volunteer's HR information was placed.** The same interpretation follows for the other volunteers.

prediabetes. It is seen that all volunteers had group variation in each cluster, this indicates 100% variability. For the control volunteers, one notices less variability. That is, of the 25 volunteers, 16 had variation between groups, which corresponds to 64% of the total.

Figure 6B shows the variability percentages for other clusters numbers.

### RR×HR analysis

Next, one has a cluster analysis of the RR *vs.* HR data for control and volunteers with prediabetes. Figure 6C shows the results of this analysis. For example, for a number of six clusters, it was found that for minutes 11, 12 and 13 the index of variability is 100% for people with prediabetes and 76% for control patients.

By observing Figs. 6A–6C, it can be identified that there is a greater distance in the curves between values 5 to 7. This distance indicates the relative variability between the two groups, as well as, the point of greatest separation between them. Thus, it is established as six the number of K adequate (of the K-means algorithm). This value is in accordance with the one obtained by the WCSS algorithm.

### Clusters names

As observed in the previous sections, the clusters follow certain number/color patterns. Thus, one can name these clusters according to their pattern as per Fig. 7. One can create a

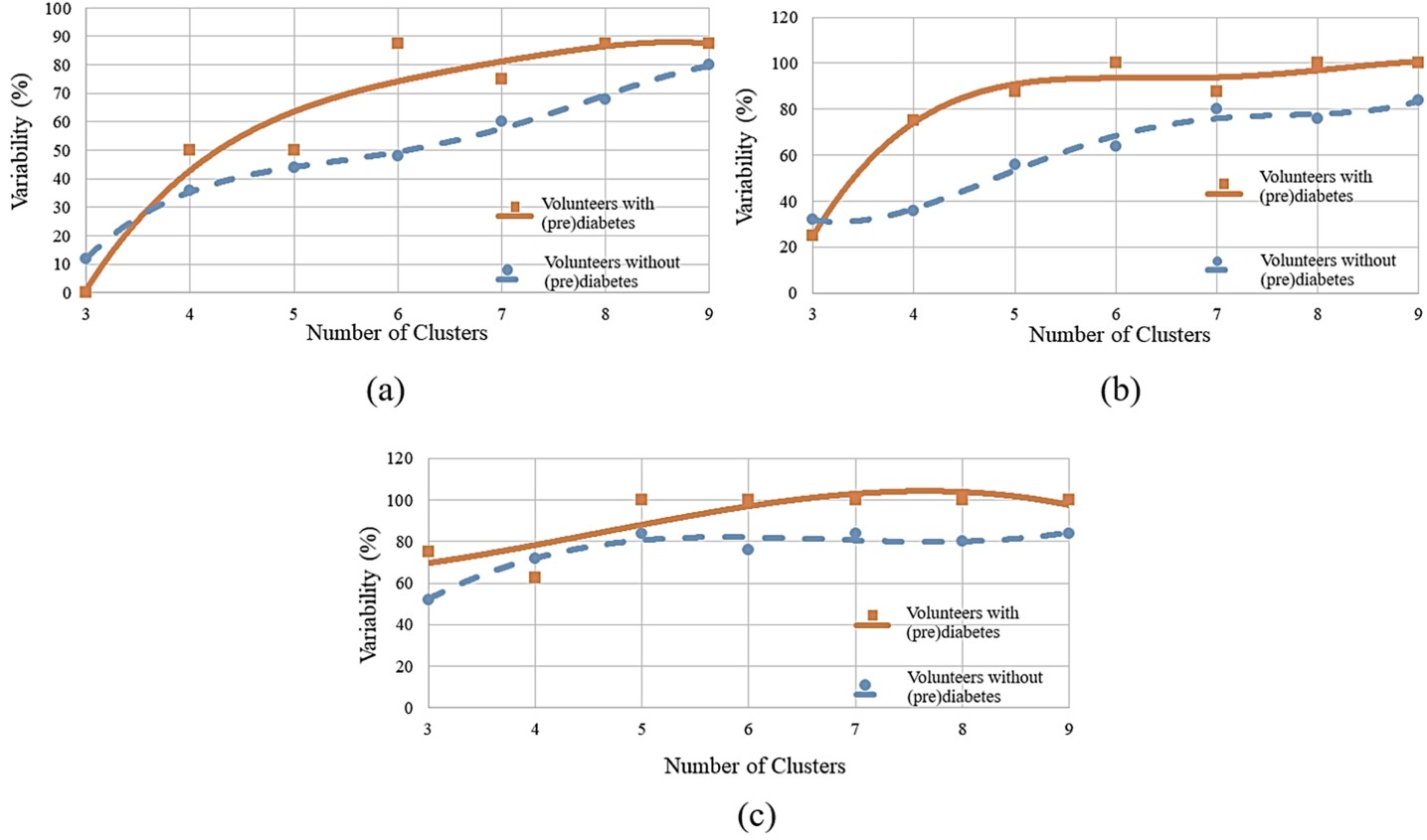

**Figure 6** Variability for different numbers of clusters: (A) HR data; (B) RR data; (C) RR×HR data.

function to convert the data in the matrix *CPOX* into a matrix with the cluster names, attributing values to them. This is the idea of the Convert of Algorithms 2 and 3.

In Fig. 8, for the K parameter of the K-means algorithm equal to 6, the variabilities for the HR, RR and RR×HR variables of the volunteers with prediabetes and without prediabetes are compared. Note in the Fig. 8 (generated from the data in Fig. 6), and according to the variability Formula (1), that the clusters of different colors (DD, DUD, DU and T) represent the value of variability for the different variables. In these graphs, one can clearly see a different quantification of clusters between the control and the volunteers with prediabetes. This quantitative relationship enables the creation of a score in order to rank the volunteers with prediabetes, their clusters and the glucose data collected. In the Score and maximum risk subsection, the clusters are quantified and differentiated according to their different types. From the establishment of this score, one can obtain a risk classification for other patients through their clusters.

## Training process: score matrix and maximum risk number

In this second part of the results, one presents the training process of the PRCA, which is systematized by Algorithm 2. After the entire clustering process performed by Algorithm 1 using K-means, Algorithm 2 takes as input the clusters of the volunteers with prediabetes

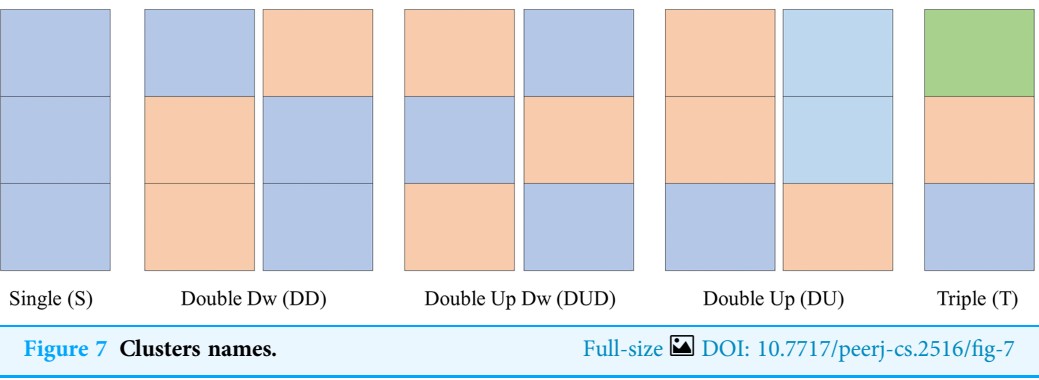

**Figure 7 Clusters names.**

**Figure 8 Graph of comparison of variability with K = 6.**

and their glucose values. As a result, the score matrix and the maximum risk number are generated.

Firstly, the association matrix $\Delta$ (Table 3) must be obtained, where the samples used for training, the weights derived from their glucose values, and the clusters provided by Algorithm 1 are related. The weights are obtained by quantifying the severity level (SL), which is calculated based on the glucose levels of volunteers with prediabetes. Figure 9 depicts a graph of the glucose level of people with prediabetes and control volunteers.

The graph in Fig. 9 was used to establish a parameter to measure the SL of each volunteer with prediabetes, where the red vertical lines are the time markers that represent the disconnection of the means of the volunteers with prediabetes and the control volunteers. Then the SL of each volunteer, calculated by Eq. (2), is obtained by the mean glucose outside the interval $(t_g, T_g)$ (border markers of separation of glucose data from the means of volunteers with prediabetes–see Fig. 9), where $g_{i,j}$ is the glucose level (mg/dl) of volunteer $i$ at time $j$.

$$SL_i = \frac{\sum_{j=1}^{t_g} g_{i,j}}{t_g} + \frac{\sum_{j=T_g}^{80} g_{i,j}}{T_g - t_g}, \text{ for } i = 1, \ldots, 8, \text{ and } j = 1, \ldots 80. \tag{2}$$

**Table 3 Matrix Δ: Weights and clusters of each volunteer with prediabetes, where W is the weight calculated for each patient from Eqs. (2) and (3).**

| Patient ID | Weight W | Clusters HR | Clusters RR | Clusters RR×HR |
|------------|----------|-------------|-------------|----------------|
| 20180101001 | 2.1815 | DU | DUD | T |
| 20180101002 | 2.8356 | DU | DD | DUD |
| 20180101003 | 2.476 | DU | DD | T |
| 20180101004 | 2.175 | S | DUD | DUD |
| 20180101005 | 1.5246 | DUD | DD | DUD |
| 20180101006 | 1.6958 | DD | DU | T |
| 20180101007 | 2.1543 | DU | DUD | DU |
| 20180101008 | 2.8676 | DD | DUD | DUD |

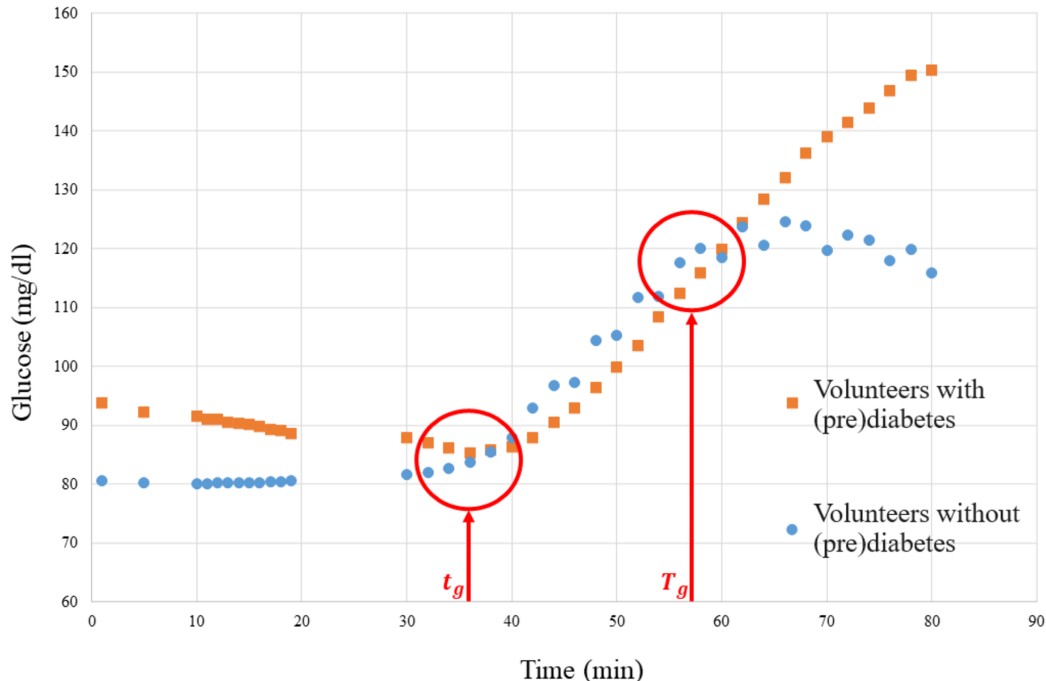

**Figure 9 Graph of the media glucose of the volunteers.**

From the SL, according to Eq. (3), a normalized weight ($W$) between 1 and 10 can be established to be used to obtain the score.

$$W_i = \frac{SL_i}{100}.$$
$$(3)$$

For example, the weight ($W$) of the patient 20180101001 is calculated as follows:
1. As shown in Fig. 9, take $t_g = 34$ and $T_g = 55$, and consider for this patient $i = 1$;
2. Apply Eq. (2) using the glucose data collected from this patient, then:

$$SL_1 = \frac{\sum_{j=1}^{34} g_{i,j}}{34} + \frac{\sum_{j=55}^{80} g_{i,j}}{55 - 34} = 93 + 125.15 = 218.15;$$

**Table 4 Score matrix ($\Psi$), where the values represent the sum of each cluster name for each variable, calculated from Table 3.**

| Cluster name | Score HR | Score RR | Score RR×HR |
|---|---|---|---|
| DU | 9.64 | 1.69 | 2.15 |
| DD | 4.56 | 6.83 | 0 |
| DUD | 1.52 | 9.37 | 9.4 |
| S | 2.175 | 0 | 0 |
| T | 0 | 0 | 6.35 |

3. Apply Eq. (3):

$$W_1 = \frac{SL_1}{100} = \frac{218.15}{100} = 2.1815.$$

The volunteers with prediabetes were organized according to their ($W$) and associated with their clusters within each variable. For this, the association matrix $\Delta$ was defined, where the rows represent each volunteer with prediabetes. The first column represents the associated weight and the other columns the clusters of the variables HR, RR and RR×HR. Table 3 shows this representation.

Finally, to construct the score matrix ($\Psi$) one must take the cluster names as rows and the variables as columns, in order to obtain a matrix $\Psi \in \mathbb{R}^{n \times m}$ where $n$ is the number of cluster types (names) and $m$ is the number of variables. Each term of the matrix ($\Psi_{i,j}$) is calculated from the matrix $\Delta$ (Table 3). The calculation is performed by adding the weights of the clusters of the same name that appear within a given variable. For example, cluster DU appears four times in variable HR in Table 3, being its weights 2.1543, 2.1815, 2.476 and 2.8356 respectively. Then the sum of these weights provides the term $\Psi_{1,1} = 9.64$. Table 4 gives the values obtained for all the terms of the $\Psi$ matrix.

Given the Table 4, it is observed that the maximum score that is possible to obtain for the whole $\Psi$ matrix is $\chi = 28.41$, considering the maximum value of each of the variables. In general, this result can be obtained by the expression 4:

$$\chi = \sum_{j=1}^{m} \max_{i} \{\Psi_{i,j}\}. \tag{4}$$

Algorithm 2 establishes all the procedures for constructing the $\Delta$, $\Psi$ matrices and obtaining the number $\chi$.

## Prediabetes risk classification algorithm

The third and final part of the PRCA is characterized by Algorithm 3, whose function is to indicate the level of risk in any given subject. To do this, Algorithm 3 uses the clustering process of Algorithm 1 and the training information provided by Algorithm 2 (score matrix and maximum risk number). Algorithm 1 is used to obtain clusters of HR, RR and RR×HR data for the individuals under analysis, who were submitted to the CBmeter

**Table 5 Risk table for prediabetes, where the range values are calculated from the maximum risk established by Eq. (4).**

| Range | Risk |
|---|---|
| 0 ⊢ 9.47 | Low |
| 9.47 ⊢ 18.94 | Medium |
| 18.94 ⊢ 28.41 | High |

protocol (which includes inhaling $O_2$ and eating a specific meal). From this, Algorithm 3 calculates the individual's score (from the Score Matrix, adding up the value of each cluster, *i.e.*, HR, RR and RR×HR of the subject under analysis) and then indicates the risk by checking which of the ranges in Table 5 their score falls into. Note that the process in Algorithm 3 is not invasive, as the algorithm is already trained.

This table is built according to the maximum risk number ($\chi$), which is divided into three parts, where the first part indicates the score range for low risk; the second part, for medium risk; and the third part, for high risk. Next, Algorithm 3 is presented, which establishes the procedures for classification.

Two examples are given. These examples illustrate how the risk classification algorithm for prediabetes works. The objective is to verify what the risk level is for a given patient to enter the group of people with prediabetes from the HR, RR, RR×HR clusters generated by the K-means algorithm. Furthermore, this example compares two volunteers from the control group and correlates the level of risk found with the level of severity *via* glucose (SL) calculated from Eq. (2) of the respective volunteer.

### Control volunteer 20180102001

The HR, RR and RR×HR clusters generated for this volunteer are shown in Fig. 10A. In the same figure, at the bottom, the score is associated according to Table 4 for each variable (HR, RR and RR×HR). Summing the score for each variable, the value 9.005 is obtained, which fits a low risk level for this volunteer to enter the group of people with prediabetes. For this volunteer, the SL is equal to 180.25.

### Control volunteer 20180102002

The HR, RR and RR×HR clusters generated for this patient are shown in Fig. 10B, being all the same type (double up). Summing the score for each variable, the value 13.48 is obtained, which fits a medium risk level for this patient to join the group of people with prediabetes. For this patient, the SL is 191.81.

### Performance and validation

The algorithm's performance can be seen from the graph in Fig. 11. This graph shows the trend in the risk of control volunteers (blue dashed line with circular markers) after being subjected to the algorithm. It can be seen that as blood glucose increases (SL), the risk also increases, but to a more moderate extent compared to volunteers with prediabetes (orange

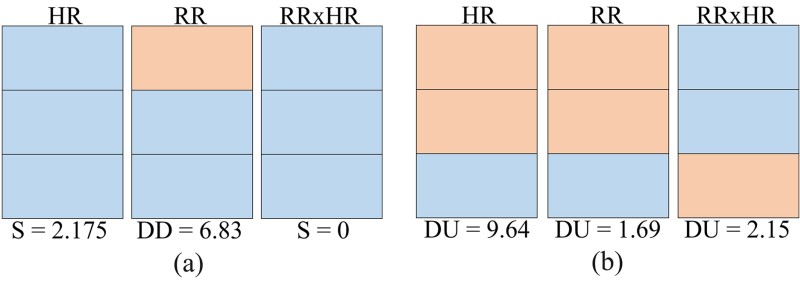

**Figure 10 Risk test for two volunteers: (A) volunteer 20180102001; (B) volunteer 20180102002.**

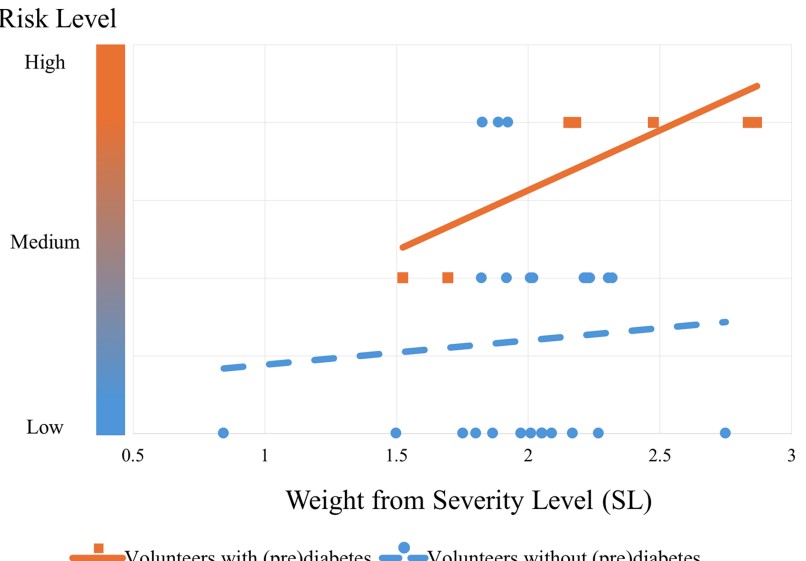

**Figure 11 Risk trend of control volunteers entering the group of people with prediabetes.** The graph shows the risk of the volunteers being in the prediabetes group: the lower bubbles represent volunteers with low risk (they do not actually have prediabetes); the bubbles and squares in the middle represent medium risk (some have prediabetes and some do not); and the bubbles and squares at the top represent a high risk of being in the prediabetes group (the orange squares are actually volunteers with prediabetes and the blue bubbles by the standard test do not currently have the disease, but according to the algorithm they are at risk of it).

continuous line with square markers). The trend lines were obtained by simple linear regression from the markers that were classified by the PRCA Algorithm.

**Remark 3.** *Note that Fig. 11 shows the performance of the algorithm trained with all the volunteers with prediabetes. This is different from validation. The cross-validation technique was used for validation, which will be explained below.*

At this stage of the CBmeter investigation, the validation of the PRCA can be approached by treating it as an artificial intelligence classification task. In this case, volunteers labeled as High Risk are considered to have prediabetes, while those labeled as Low Risk are considered to be without prediabetes. For simplification, volunteers labeled as Medium Risk are regarded as successful classifications.

**Table 6 Cross-validation process four-fold: in bold, test samples; and without bold, training samples.**

|  | Group 1 | Group 2 | Group 3 | Group 4 |
|---|---|---|---|---|
| | **20180101008** | 20180101002 | 20180101005 | 20180101007 |
| Round 1 | **20180101001** | 20180101006 | 20180101004 | 20180101003 |
| | **Test** | Training | Training | Training |
| | 20180101008 | **20180101002** | 20180101005 | 20180101007 |
| Round 2 | 20180101001 | **20180101006** | 20180101004 | 20180101003 |
| | Training | **Test** | Training | Training |
| | 20180101008 | 20180101002 | **20180101005** | 20180101007 |
| Round 3 | 20180101001 | 20180101006 | **20180101004** | 20180101003 |
| | Training | Training | **Test** | Training |
| | 20180101008 | 20180101002 | 20180101005 | **20180101007** |
| Round 4 | 20180101001 | 20180101006 | 20180101004 | **20180101003** |
| | Training | Training | Training | **Test** |

The validation was carried out using the four-fold cross-validation technique (see more in *Wong & Yeh (2019)*). This means the volunteers with prediabetes were randomly divided into four groups, each containing two samples, resulting in a total of eight samples. Four rounds of testing were performed, with each round using one group for testing (two samples) and the remaining three groups (six samples) for training. Table 6 illustrates this process. The proportion for testing and training is similar to that used by *Ahmed et al. (2022)* of 70:30, *i.e.*, 70% of the data for training and 30% of the data for validation.

The training process involves using the volunteers with prediabetes to determine the maximum risk number ($\chi$) in each round. Since the PRCA algorithm employs unsupervised learning *via* k-means clustering, it groups data based on distance measures, forming distinct clusters. Although the algorithm is trained only with data from volunteers with prediabetes, it can also classify synthetic data (generated from the volunteers with prediabetes) and control group data. This is possible because PRCA identifies patterns within the data and separates groups based on their proximity or distance from the training data. Generally, the farther a patient's clusters are from the training data clusters, the lower their risk of having prediabetes.

In the validation approach presented in this study, control group data and synthetic data generated using the ADASYN technique were added. ADASYN increased the number of data points for volunteers with prediabetes to match the number of control group volunteers. Since there are eight volunteers with prediabetes and 25 volunteers without prediabetes, the algorithm generated 17 synthetic samples from the volunteers with prediabetes.

As a result, there are 17 artificial samples from volunteers with prediabetes, plus 2 original data samples, totaling 19 samples per round. For the volunteers without prediabetes, 25 samples are available. Of these, 19 are randomly selected for each round, balancing the dataset for validation. The final validation result is the average of the performance metrics across the four rounds.

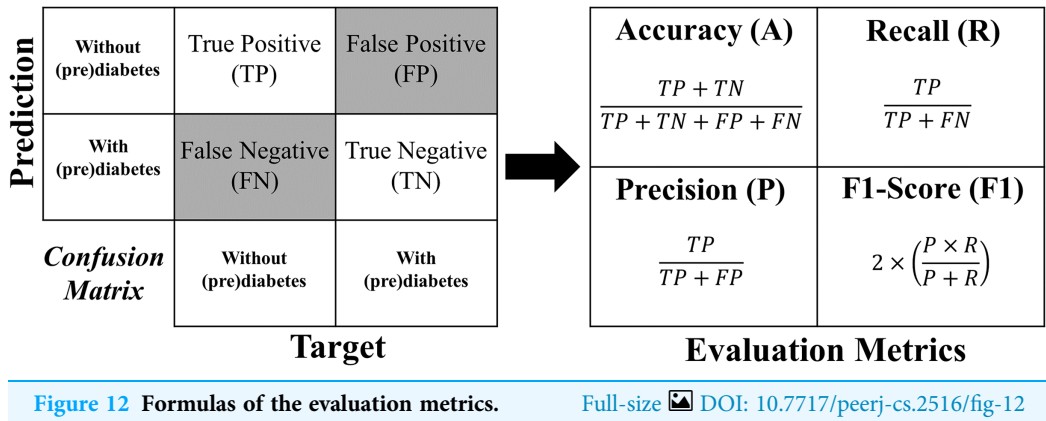

Figure 12 Formulas of the evaluation metrics.           

The metrics used for evaluation are accuracy, precision, recall, and F1-Score, which are described in more detail below:

- **Accuracy** shows the overall performance of the model, indicating, among all diagnostics, how much the model indicated correctly. Accuracy is a good overall indication of how the model performed. However, there may be situations in which it is misleading, so it is necessary to check the other metrics as well.
- **Precision** checks how many of the positive classifications the model has made are correct. Precision can be used in a situation where false positives are considered more harmful than false negatives.
- **Recall/Revocation/Sensitivity** calculates among all the positive class situations as expected value, how many are correct. The recall can be used in a situation where the false negative is considered more harmful than the false positive.
- **F1-Score** is the harmonic mean between precision and recall. The F1-Score is simply a way of looking at only one metric instead of two (precision and recall) in a given situation. It is a harmonic mean between the two, meaning that when you have a low F1-Score, it is an indication that either precision or recall is low.

The formula for each metric is obtained from the confusion matrix structure shown in Fig. 12.

Figure 13 shows the confusion matrix for the four rounds of cross-validation. Subsequently, according to the formulas in Fig 12, $A$, $P$, $R$ and F1-Score ($F1$) are calculated for each round, with the final result being the simple arithmetic mean of each metric. Thus, using the four-fold cross-validation technique, an accuracy of 86%, precision of 95%, recall of 78%, and F1-Score of 85% are obtained.

## Comparison with other methods

Considering the PRCA as a classifier, it can be compared with other classification algorithms. Using MATLAB® R2020a and the same dataset, *Pinheiro, Guarino & Fonseca-Pinto (2024)* developed a classification algorithm *via* support vector machine (SVM) with linear, polynomial, and RBF kernels. The same data balancing method (ADASYN) was

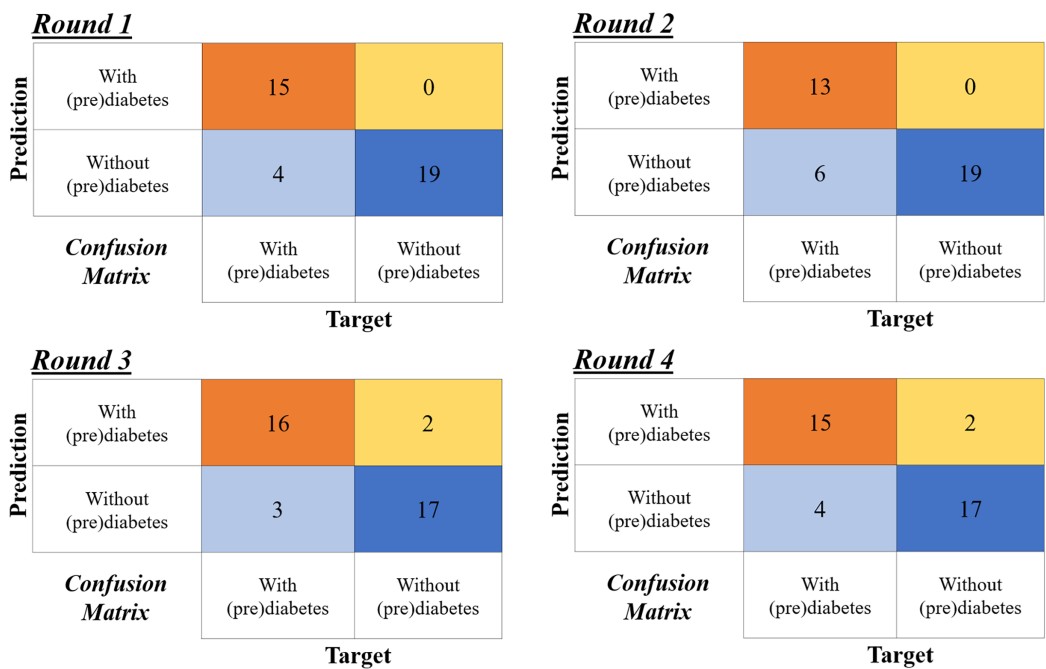

**Figure 13 Confusion matrices of the four-fold cross validation according to the rounds in Table 6.**

**Table 7 Comparison of PRCA with SVM methods by *Pinheiro, Guarino & Fonseca-Pinto (2024)* with same dataset.**

| Method | Accuracy (%) |
|---|---|
| PRCA | 86 |
| SVM Polynomial | 82.4 |
| SVM Linear | 78.6 |
| SVM RBF | 77.4 |

used. In that work, the results were obtained as shown in Table 7. Upon examining the table, it is observed that the PRCA has better accuracy.

Although a reasonable comparison for PRCA is presented in Table 7, for a better overview of what is available in the literature, Table 8 presents some other studies aimed at supporting diabetes diagnosis. It is important to note that the Table 8 should not be used to directly compare performance with PRCA, as the studies listed were conducted under substantially different conditions, using different datasets, methodological approaches and parameters from those applied in the development of PRCA. The table includes information on the method, sample size, number of resources, accuracy and whether the method is invasive or non-invasive. The table shows that all algorithms use a greater number of features and the vast majority of them are invasive. In addition, they all have a larger number of samples, which is a limitation of this study.

Notably, in addition to being a non-invasive method, PRCA uses fewer features, which indicates a lower computational cost. Moreover, with the acquisition of additional samples

**Table 8 Other methods and datasets in the literature.**

| Work | Method | Sample | Features | Accuracy (%) | Invasive |
|---|---|---|---|---|---|
| *Khan et al. (2024)* | ANN + feature selection | 768 | 6 | 99 | Yes |
| *Ahmed et al. (2022)* | SVM + ANN + Fuzzy | 520 | 17 | 94 | No |
| *Patil et al. (2022)* | Modified mayfly-SVM | 1,133 | 33 | 94 | Yes |
| *Zhu, Idemudia & Feng (2019)* | K-means + Log. Reg. + PCA | 1,500 | 13 | 89 | Yes |
| PRCA (this work) | CB + K-means + Score | 33 | 3 | 86 | No |
| *Zhu, Idemudia & Feng (2019)* | K-means + XGBoost + PCA | 1,500 | 13 | 85 | Yes |
| *Nilashi et al. (2023)* | ANN + Self-Organizing Map | 768 | 6 | 84 | Yes |
| *Ramesh, Aburukba & Sagahyroon (2021)* | SVM | 768 | 6 | 83 | Yes |
| *Zhu, Idemudia & Feng (2019)* | K-means + Naïf Bayes + PCA | 1,500 | 13 | 82 | Yes |
| *Zhu, Idemudia & Feng (2019)* | K-means+ KNN + PCA | 1,500 | 13 | 78 | Yes |
| *Zhu, Idemudia & Feng (2019)* | K-means + SVM + PCA | 1,500 | 13 | 58 | Yes |

in the continuation of this investigation in the future, the algorithm's performance could improve, as a larger dataset provides more examples for the model to learn from (*Elfatimi, Eryiğit & Elfatimi, 2024*).

## DISCUSSION

In the construction stage of the PRCA, interstitial glucose measurements, which are minimally invasive, were necessary for training purposes. However, once the algorithm is fully trained and operational, it no longer requires interstitial glucose data. Instead, it relies solely on HR and RR data, which are influenced by oxygen and meal stimuli. As a result, the continuous application of this algorithm does not require invasive procedures, thus designating it as a non-invasive method for monitoring and classifying the risk of prediabetes.

In the development of the clustering technique, the clusters were performed using the K-means algorithm with different values of K in order to obtain the best parameter. Besides, the WCSS algorithm was used, confirming that the adequate number of clusters was six for the three feature variables (HR, RR and RR×HR) from a simple average. However, the algorithm could have also been developed by establishing the number of clusters for each feature. It was not done in order to reduce the complexity of the presented issue.

In this work, good results were obtained using a frequency down-sampled to 1-min bins for HR and RR. This fact shows the power of the algorithm with respect to computational cost in relation to its accuracy (86%), because the higher the frequency of analysis, the greater the amount of data, thus increasing the computational cost. However, the algorithm is easily adaptable to work with higher frequencies, a fact that would increase the computational cost, although it may have a better screening of prediabetes or for other types of diseases in which it can be applied. In future work, with a larger number of volunteers, the intention is to analyze in detail the issue of the cost-benefit of the method, with regard to the number of volunteers, as well as the frequency used.

Considering PRCA as a classification algorithm, it was possible to make a comparison with the results of another classifier using the SVM method (see Table 7), which has a completely different approach in relation to the algorithm developed in this work. SVM is a powerful tool in the field of machine learning for building algorithms that help diagnose diseases. SVM is based on supervised learning, while the PRCA method uses an unsupervised learning technique. The comparison shows that the best classifier using the SVM technique achieved an accuracy of 82.4%, ranking behind PRCA, which achieved an accuracy of 86%. Additional considerations are made on the basis of Table 8, highlighting PRCA as one of the few methods with a non-invasive approach, as well as achieving good results using few features and a reduced number of samples. Another important point is that PRCA is the only method, among those shown in the table, that offers a strategy aimed specifically at screening for prediabetes. Among the methodologies that use the K-means algorithm, PRCA stands out positively compared to the others. It is important to note that among these methods, only PRCA adopts a non-invasive approach.

The results of the cluster analyses, made it possible to verify some mathematical topological properties that indicate the separability of the data found after the clusters. For example, Figs. 3, 6, 8 and 9 in all the points of prediabetes and control it is possible to find a disjoint neighborhood that separates from other points. Being at the extremes, this separation greater. In mathematics, this type of topology can be within the Hausdorff spaces that contain important properties to develop algorithms using machine learning and make statistical inferences (*Chen, Genovese & Wasserman, 2017*; *Dasoulas et al., 2019*).

There are many works in the literature that use score techniques in the design of algorithms to determine the risk for some pathology. The technique of scores and determination of a maximum risk value developed in this work is an alternative heuristic of low computational cost carefully designed to associate the clusters and diabetes to classify the risk. With the approach used, the algorithm becomes of easy change for adjustments and validations in other pathologies.

The concept of severity level (SL) was elaborated from the glucose level (fasting and postprandial) of the volunteers submitted to the CBmeter protocol, that is, assessing changes in HR, RR, SpO2 and plasma glucose after 100%O2 administration and after eating a standardized meal. In contrast to the gold standard, this methodology submits the patient to conditions that better mimic their daily life, making it possible to detect blind spots that the gold standard does not capture. Moreover, compared to fasting blood glucose tests, the CBmeter detects metabolic dysfunctions earlier, before fasting glucose rises, because the CBmeter offers a more detailed analysis of the sensitivity of the carotid body, allowing early identification of insulin resistance (*Conde et al., 2022*). Although it has a longer evaluation time, its duration is similar to the OGTT. However, further research is needed to systematically validate the SL with this algorithm, to include more parameters in addition to glucose (fasting and postprandial). For example, fasting insulin could be included as an additional parameter for SL. Furthermore, more research is needed to make better comparisons with the gold standard test, which is one of the limitations of this study.

Another limitation of this study is the sample size. The study considered eight volunteers with prediabetes and 25 volunteers without prediabetes. Although it is well known that small samples negatively affect the quality of group-based estimates, this work presents an algorithm that, according to preliminary results, appears promising. This encourages continued investigation for effective validation and improvement of the algorithm's quality, aiming for future implementation in the medical diagnostic equipment industry.

An important issue to be discussed, it is if in the future the algorithm presented in this work could be considered a disease predictor algorithm and not just a classification algorithm. The predictive capacity would be inherent in the developed CB method and in the process of constructing and training the algorithm. This is explained by the fact that the algorithm is built on an intervention that stimulates the CB, where this response to stimuli is considered in the literature to be a biomarker for detecting prediabetes. As mentioned in the introduction, a biomarker is a measurable biological characteristic that indicates an ongoing pathogenic process, and the conclusion of this process will be (in the future) the natural characterization of the disease. In addition, the PRCA training process is carried out only with the eight volunteers with prediabetes, and the validation shows its ability to determine the risk for people without prediabetes. However, in order to really say that the algorithm presented in this work is a predictor, it would be necessary to continue the investigation and validate it, which requires a longitudinal study of the participants.

Therefore, it should be noted that the PRCA is an initial prototype of an ongoing investigation, with the ultimate goal being its definitive validation as a predictor. A long time is required to know if a certain volunteer who was classified with a high level of risk, developed the disease or not. Thus, the true validation of a possible risk predictor algorithm based on the methodology presented in this article is a complex process that requires a longitudinal study of the participants.

## CONCLUSION

This article presents a potential prediabetes risk classification algorithm (PRCA) based on unsupervised machine learning technique from real CBmeter data. The proposal of the algorithm is ambitious, as it aims to identify possible patients with prediabetes who are not captured by the current gold standard methods. It was shown throughout the work that the algorithm is promising, taking into account its creation methodology and its performance. This article is considered a pilot study for the development of a future risk predictor algorithm. On the other hand, in a simpler view, PRCA can also be considered as a classifier, which proved to perform better than the classifiers developed using the SVM method.

PRCA, developed under the CBmeter protocol, makes a significant contribution by providing the possibility of detecting insulin resistance early in a non-invasive way, before fasting blood glucose changes. Unlike traditional tests that require blood sampling, it uses ventilatory and cardiac responses to hyperoxia to identify prediabetes. This approach can

simplify diagnosis, improve the patient experience and facilitate early screening, complementing or even surpassing invasive methods such as fasting glucose. In addition, PRCA has the ability to indicate the risk of prediabetes using a methodology that more accurately reflects people's usual eating patterns, while also avoiding the collateral effects often associated with OGTT, such as vomiting, diarrhea, bloating, hypoglycemia and other possible complications.

Future work will continue the CBmeter investigations to achieve precise validation through complementary longitudinal studies of the volunteers, monitoring who develops the disease, and conducting studies with a larger number of volunteers. Furthermore, it is desired to adapt the algorithm developed in this article to other metabolic diseases so that, in the future, the CBmeter will reach its implementation phase (TRL8) for the medical device industry.

### Funding
This work was funded by Portuguese national funds provided by Fundação para a Ciência e Tecnologia: FCT-UIDB/05704/2020 and CEECINST/00051/2018 regarding Maria P. Guarino collaboration; and in the scope of the research project 2 ARTs—Acessing Autonomic Control in Cardiac Rehabilitation (PTDC/EMD-EMD/6588/2020) co-financed by the Portuguese Foundation for Science and Technology (FCT). Rafael Pinheiro received financial support from the FCT through the Institutional Scientific Employment Stimulus CEECINST/00060/2021. The funders had no role in study design, data collection and analysis, decision to publish, or preparation of the manuscript.

### Grant Disclosures
The following grant information was disclosed by the authors:
Portuguese Foundation for Science and Technology (FCT): FCT-UIDB/05704/2020, CEECINST/00051/2018.
Acessing Autonomic Control in Cardiac Rehabilitation: PTDC/EMD-EMD/6588/2020.
FCT through the Institutional Scientific Employment Stimulus: CEECINST/00060/2021.

### Competing Interests
The authors declare that they have no competing interests.

### Author Contributions
- Rafael F. Pinheiro analyzed the data, performed the computation work, prepared figures and/or tables, authored or reviewed drafts of the article, and approved the final draft.
- Maria P. Guarino conceived and designed the experiments, performed the experiments, authored or reviewed drafts of the article, and approved the final draft.
- Marlene Lages conceived and designed the experiments, performed the experiments, authored or reviewed drafts of the article, and approved the final draft.
- Rui Fonseca-Pinto conceived and designed the experiments, analyzed the data, authored or reviewed drafts of the article, and approved the final draft.

## Ethics

The following information was supplied relating to ethical approvals (*i.e.*, approving body and any reference numbers):

The study was conducted in accordance with the Declaration of Helsinki, being previously approved by the Ethics Committee of the Leiria Hospital Centre (Protocol number PI.NC.EC.2018.01). Informed consent was obtained from all participants and/or their legal guardians.

## Data Availability

The raw data is available in the Supplemental File.

## Supplemental Information

Supplemental information for this article can be found online at http://dx.doi.org/10.7717/peerj-cs.2516#supplemental-information.

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
