# Peer review of "Prediabetes risk classification algorithm via carotid bodies and K-means clustering technique"

_PeerJ Computer Science, doi:10.7717/peerj-cs.2516_

## Round 0.1 · original submission · Major Revisions

The three reviewers raised a number of issues that make this article insufficient for acceptance and publication. Please address all the points indicated by the reviewers and prepare a new version.

·

Basic reporting

I read the manuscript interestingly. Although the manuscript is presented well, I have several reservations:
The sample size is small. Small samples negatively affect the quality of the information we use when making group-based estimates.
The writing is too colloquial and needs a fluent English speaker.
The discussion should be more elaborative in line with the results.

Experimental design

Experimental design is good.

Validity of the findings

The belief that results from small samples are representative of the overall population is a cognitive bias.

Reviewer 2 ·

Basic reporting

The paper proposes an algorithm for predicting prediabetes risk using carotid body measurements and the k-means clustering technique. Overall, the paper is easy to follow, and the context and the developed method are straightforward. Although English is not my native language, the English used is professional and understandable. The paper addresses a very important and relevant problem in the international community. However, I raise some critical points:

Specific Comments:

1. Introduction: While the introduction is appropriate, it would be greatly improved by providing more details on machine and deep learning architectures for the diabetes risk problem (even if not necessarily based on carotid body measurements).

2. Figures: The figures are relevant; however, all figures need more detailed captions (e.g., title, a brief description of the figure, what the x and y axes are, and the meaning of colors, especially in figures 6 and 7, 14 and 15). The number of figures can be reduced; for example, figures 14 and 15 can be combined.

3. Data and Code: The raw data and code are provided. It is recommended that details be added on executing the individual parts to replicate all results exactly and allow the model to be used in inference. This is of fundamental importance to the international community.

Experimental design

4. Participant Selection and Data Characteristics:
The "Participants and Data Collection" section mentions 8 volunteers with prediabetes or diabetes and 25 healthy controls. However, the study lacks details on the selection process for these participants, which could introduce selection bias into the experiment.

The paper fails to adequately emphasize the dataset's dimensionality. It is crucial to highlight that the dataset is not two-dimensional (variables X patients) but three-dimensional (variables X patients X time). The authors should consider using an appropriate formalism to convey this concept effectively.

5. Clustering Algorithm Selection and Justification:
The discussion on k-means clustering is satisfactory and effectively determines the optimal number of clusters. However, given the availability of various state-of-the-art clustering algorithms, the paper should include a comparison with alternative methods or provide a more thorough justification for choosing k-means.

6. Performance Evaluation and Generalizability:
I am concerned about the performance evaluation and validation methodology in line 291. If I understand correctly, the test set comprises the same patients used to establish the k-means model. Have the authors performed data augmentation and tested the model on augmented data? The paper should address the potential for overfitting due to the use of the same data for both model training and evaluation. Employing a separate validation dataset or implementing cross-validation techniques would be more appropriate.

Validity of the findings

7. Absence of a True Validation Dataset:
The paper lacks a dedicated validation dataset for evaluating the model's performance on unseen data. This omission raises concerns about the model's generalizability and potential for overfitting.

8. Small Dataset Size and Inappropriate Metric:
The limited size of the dataset, with only 8 volunteers with prediabetes or diabetes and 25 healthy controls, further restricts the reliability of the results. Additionally, the use of accuracy as the primary evaluation metric is inappropriate for imbalanced datasets like this one. Alternative metrics such as F1-score or precision-recall curves should be employed to provide a more comprehensive assessment of the model's performance.

Additional comments

Minor comments:
- Modify Figure 5 to display the time interval over which the regression line is calculated on the side separately. This will help avoid confusion.

- Correct the misspelling of "patient" as "pacient" throughout the paper and in the algorithm descriptions.

Overall, this is a good paper that makes a valuable contribution to prediabetes risk prediction. However, the authors could improve the paper by addressing the abovementioned critical points.

Reviewer 3 ·

Basic reporting

See additional comments.

Experimental design

See additional comments.

Validity of the findings

See additional comments.

Additional comments

1. The authors claim that OGTT is the gold standard for diabetes diagnosis, which is true, but they ignore the reality of clinical practice, where, following ADA guidelines the authors conveniently leave out, diabetes and pre-diabetes can be diagnosed via, e.g., repeated measurements (a few weeks apart) of fasting glucose. Why is that?
2. They claim that their CB-based methodology is non-invasive. So how do they get interstitial glucose measurements?
3. I would venture that what they are doing is classifying people with and without (pre-)diabetes, and not, strictly, predicting anything (as in determining future risk). I would highlight this whenever possible, and try and avoid language that might be misunderstood.
4. Speaking of language, we prefer the term "people/individuals with diabetes;" "diabetic,"”as an adjective, should not be used to describe people.
5. Calling the matrices the authors introduce "new concepts" seems like a stretch: they are matrices of features (+ some linear combination thereof). Where would they say the novelty is?
6. The study population is very small; too small for any study that tries to establish the validity of a new algorithm, which seems to be the case here. A new clinically relevant result might be found with these numbers, but in a completely different experimental setting.
7. Looking at figure 5, I cannot help but feel that RR would be a good predictor on its own, without the experimental setup the authors develop. What is the advantage of the latter, then?
8. I am not sure if I have understood correctly, but it looks like they clustered 8 people with (pre-)diabetes into 6 clusters. I would suggest the authors explain the purpose of this clustering better.
9. The tables are difficult to follow: there is no caption describing what the numbers are.
10. What is the mathematical formula for "variability?" What is it measuring?
11. Why are the labels on the X axis of figure 12 not always the same? Would that have changed the nice regression lines? Also, why does it make sense to order the clusters like that?
12. I am not sure if any proper validation following machine learning best practices was carried out (hold-out set? Cross-validation? Bootstrap?). Would the authors clarify?

---

## Round 0.2 · Major Revisions

Some issues were solved by the authors but some others were not.

The authors should take into consideration ALL the comments of all the reviewers, and address them carefully.

Reviewer 2 ·

Basic reporting

The authors have demonstrated a commendable effort and dedication in revising the manuscript. The current version is significantly improved and more comprehensive. I thank the authors for their thoughtful responses to the points raised in the previous review.

However, some crucial issues related to the conclusions and claims made in the manuscript still need to be addressed before publication. Specifically, the procedural aspects related to the validation set require further clarification.

Experimental design

The authors present a compelling argument for using CD as a potential biomarker for diabetes. The developed model reported accuracy/precision and recall of 88%. A more detailed comparison with existing methods is needed to assess these findings' novelty and significance fully. Compared to the state of the art (even if without the use of CD), is 88% higher or lower? This comparison can be done by searching the literature without reimplementing other works, and would greatly benefit the understanding of the problem.

Validity of the findings

I appreciate the authors' acknowledgment of the limited sample size. However, I believe there is a fundamental issue with evaluating the results. In Remark 2, the authors state that they trained their model on only 8 pre-diabetic patients. Subsequently, validation was performed on 25 non-prediabetic patients and 17 artificially augmented patients.

Firstly, the specific data augmentation technique employed should be clarified. Secondly, most of the validation set consists of augmented data, which may lead to overfitting as the augmented data will likely closely resemble the training data.

I suggest a more rigorous evaluation strategy. For instance, the 8 pre-diabetic patients could be randomly split into two sets of four. One set and its corresponding augmented data could be used for training, while the other set and its augmented data could be used for validation. Alternatively, a cross-validation approach could be employed.

Additional comments

The inclusion of detailed code replication instructions is a valuable addition to the manuscript. However, to maintain the clarity and focus of the Methods section, I suggest moving these instructions to a dedicated README.txt file. This file should be placed in the same directory as the code and provide a step-by-step guide for users who wish to reproduce the results. Additionally, consider including a brief overview of the code structure and dependencies in the README file.

Reviewer 3 ·

Basic reporting

See comments.

Experimental design

See comments.

Validity of the findings

See comments.

Additional comments

Many thanks to the authors for addressing the points I had raised concerning the original version of the manuscript. Unfortunately, as the authors may see below, I have still a number of unresolved concerns on critical matters.

1. The authors now mention the existence of much less invasive diagnostic techniques than OGTT, which is fair, but they still do not compare their method to fasting glucose assays in practical terms. For example, given that their CBMeter test takes longer than an hour, is that really preferable to a quick blood test? Does it make economic sense? Granted, a new methodology does not have to have all these requirements, but I believe it is fair to give a fair assessment of it relative to current clinical practice, and possibly highlight interesting use-cases.

2. I think I understand now that the maximum risk chi is a training-set-dependent parameter, which is estimated once, and then used as is for prediction (i.e., it’s the same number for all “validation” subjects). Is that correct? If so, that would indeed make the authors’ methodology non-invasive, and should be better highlighted in the text.

3. I appreciate the distinction between prediction and classification. The text added by the authors, however, seems like a reach. Any score could be a predictor (say, a heart disease risk score could predict diabetes, in principle), but it is unrealistic to trust a classifier to make predictions without having trained it on longitudinal data (or extensively validated it), especially in a field as risk averse as medicine. I would suggest attenuating that remark.

4. Not all instances of "diabetic" referring to people have been amended.

5. The novel aspects of the authors’ work are now better circumstantiated.

6. I maintain that the sample size the authors are considering appears insufficient for such a complex experiment. Model capacity and sample size should at least be somewhat balanced, and, most importantly (see following point), comparisons with simpler techniques (even naïve, deterministic methods) should be carried out to demonstrate that the complex methodology is actually “needed” to solve the problem. Going back to comment #1, in the current version of the manuscript it is also unclear what the “valuable contribution” to early diagnosis might be, as the authors do not show examples of use cases where fasting glucose fails while their technique works.

7. I think the authors have it backwards: when proposing a new set of predictors, the target should generally be to obtain the most parsimonious model possible (e.g., because longer or multiple assessments are expensive / time consuming / etc.). In other words, it is not so much a matter of “conclud[ing] definitively that the other variables do not contribute,” but rather of proving that any additional variable is helpful. This extends to the complexity of the algorithm: every layer of Maths that you add increases the distance between the underlying biology and the model’s output. And this is generally tolerable (as it reduces explainability) only in the face of a significant gain in performance (or other utility measure). I maintain that there is a need of extensive comparison with simpler models and reduced sets of features.

8. As for the clustering, I think I understand now that each subject switches clusters over time, or, rather, that each time point is put into a cluster. If so, the sample size for this analysis should be clarified.

9. Many of the tables remain virtually captionless. For instance, what is the “form of presentation” (sic) in Tables 1 and 2? What are the numbers? What does “59” mean? What units of measurement is it in?

10. Thanks for clarifying what variability is.

11. The new figure works better in the context of the manuscript.

12. I am not sure if I got that right, but it appears that “Data augmented from the 8 volunteers” is, basically, the training set + some noise. If so, that is in clear breach of machine learning best practices as it introduces extensive information leakage between the training and validation subsets of the data. Further explanation is warranted.

---

## Round 0.3 · accepted · Accept

The authors correctly addressed the points raised by the reviewers and therefore I consider this article acceptable for publication.

Reviewer 2 ·

Basic reporting

The authors have made substantial revisions that have rendered the article significantly more robust in terms of reproducibility. I appreciate the detailed instructions provided in the supplementary materials for replicating the results.

Experimental design

The experimental design has been improved by including more comprehensive comparisons to state-of-the-art methods, which better contextualizes the work. While I still have some reservations regarding the size of the dataset, I understand the challenges of obtaining independent third-party validation sets. Nonetheless, the authors have addressed this concern by clearly specifying the data augmentation technique used and modifying the evaluation strategy accordingly.

Validity of the findings

no comment